# Impact of Unfortified Human Milk, Fortified Human Milk, and Preterm Formula Intake on Body Composition at Term in Very Preterm Infants: Secondary Analysis of the PREMFOOD Trial

**DOI:** 10.3390/nu17081366

**Published:** 2025-04-17

**Authors:** Luke Mills, Sabita Uthaya, Neena Modi

**Affiliations:** 1Section of Neonatal Medicine, School of Public Health, Faculty of Medicine, Imperial College London, Chelsea and Westminster Hospital, 369 Fulham Road, London SW10 9NH, UK; s.uthaya@imperial.ac.uk (S.U.); n.modi@imperial.ac.uk (N.M.); 2Chelsea and Westminster NHS Foundation Trust, 369 Fulham Road, London SW10 9NH, UK

**Keywords:** anthropometry, body composition, donor milk, milk, human, preterm formula, preterm infant

## Abstract

**Background/Objectives**: Very preterm body composition at term shows potential as a biomarker of later health outcomes, but effects from in-hospital formula versus human milk (HM) (maternal milk (MM) and/or pasteurised human donor milk (DM) supplement) intake studies are confounded by the effect from the fortifier. We investigated the impact of in-hospital unfortified HM (UHM), fortified HM (FHM), and preterm formula (PTF) intake on very preterm body composition at term. **Methods**: Preplanned analysis of the PREterM FOrmula or Donor milk (PREMFOOD) trial: Infants born at <32 weeks were randomised to either (i) UHM, (ii) FHM, or (iii) MM and/or PTF supplement. Main outcomes were assessed by anthropometry and magnetic resonance imaging of body composition at term. Secondary comparison between groups defined by proportion of milk intake from birth to 35 weeks postmenstrual age: The groups comprised exclusive UHM (ExUHM, proportion of UHM 99–100%, n = 23), predominantly UHM (PrUHM, UHM 50–98.9%, n = 15), predominantly FHM (PrFHM, FHM > 50%, n = 17), and predominantly PTF (PrPTF, PTF > 50%, n = 7). **Results**: At term, compared to the ExUHM group, the PrPTF group had 274.3 g (95% CI: 30.1 to 518.5) more Non-Adipose Tissue Mass (NATM) and a 1.11 times (95% CI: 0.38 to 1.84) greater increase in weight z score from birth, while both PrPTF and PrFHM had greater increases in length z scores from birth. **Conclusions**: High formula intake was associated with maximal gains in NATM at term, and these gains were not matched by the early fortification of HM. The alteration of body composition at term with prolonged or delayed HM fortification and its relation to later health outcomes are important research questions.

## 1. Introduction

Uncertainty exists regarding the optimal supplementation of mother’s own milk (MM) for very preterm infants. Pasteurised human donor milk (DM) and preterm formula are options. However, pasteurisation reduces or destroys the non-nutrient benefits of fresh MM, and unlike with preterm formula, due to the variable nutrient content of human milk (MM and/or DM (HM)), many practitioners also provide bovine-derived multicomponent fortification with the aim of optimising growth and development.

Recent preterm feeding trials, including current meta-analyses, have found faster short-term growth with formula versus donor milk or with fortified HM (FHM) compared to unfortified HM (UHM) but with no difference in long-term anthropometry or neurodevelopment [1,2,3]. Clinical trials adequately powered for important clinical outcomes with sample sizes large enough to establish long-term follow up are required to address these uncertainties, ideally utilising early biomarkers that can predict long-term outcomes. Body composition at term may be one such candidate biomarker. The growing knowledge of increased risk of hypertension, diabetes, and the metabolic syndrome [4,5,6,7] among very preterm survivors has raised the possibility of early nutrition mediating these effects through rapid growth velocity and/or altered adiposity. Limited data from nutritional intervention studies in preterm infants do not suggest the long-term modification of absolute adiposity [8], but many studies have measured body composition using techniques unable to quantify adipose distribution. Using magnetic resonance imaging (MRI), our group previously showed that Internal Abdominal (visceral) Adipose Tissue (IAAT), a biomarker of metabolic risk in children and adults [9,10], is increased in preterm infants at term [11] and in young adult life [12] compared with individuals born full-term. Non-Adipose Tissue Mass (NATM), or fat-free mass at term, on the other hand, has been associated with higher Bayley composite motor and language scores at 2 years [13].

To date, observational studies comparing body composition at term in preterm infants have reported discordant findings in the comparison of predominant or exclusive diets of HM versus preterm formula, with some reporting higher fat mass and fat mass percent with preterm formula feeding [14,15], while others report no difference in adiposity but higher Non-Adipose Tissue Mass (NATM) [16]. Interpretation is limited, however, since none of these studies have compared all three feed exposures of UHM, FHM, and preterm formula.

As part of a feasibility trial powered to detect important clinical outcomes comparing very preterm diets of UHM, FHM, and preterm-formula-supplemented MM (PTF) until 35 weeks postmenstrual age (PMA), we previously reported that despite a significantly higher change in weight z score at 35 weeks PMA seen in the PTF group, there were no significant differences in anthropometry or MRI-measured body composition by term [17]. However, high intake of MM in all randomised groups and subsequent feed exposure from 35 weeks PMA may have blunted any effect on term anthropometry and body composition. Thus, as part of a preplanned secondary analysis and in the context of detailed nutritional intake throughout the hospital stay, our objectives here were to assess anthropometry and body composition at term dependent on the proportion of intake of human milk or preterm formula from birth to 35 weeks PMA, comparing an exclusive UHM (ExUHM, proportion of UHM 99–100%) intake group with predominant UHM (PrUHM, proportion of UHM 50–98.9%), predominant FHM (PrFHM, FHM > 50%), and predominant preterm formula (PrPTF, PTF > 50%) intake groups.

## 2. Materials and Methods

### 2.1. Study Design

We conducted PREMFOOD (PREterM FOrmula or Donor milk for preterm babies), an open, multicentre, parallel, randomised controlled feasibility trial. The trial was preregistered at ClinicalTrials.gov (NCT01686477) and approved by the UK National Research Ethics Service (REC no: 12/LO/1391). Further details of the feasibility trial protocol are reported elsewhere [17]. Briefly, preterm infants born between 25 + 0 and 31 + 6 weeks, without congenital abnormalities that precluded early milk feeding, who were unlikely to be transferred to another hospital, and who were randomised by opt-out consent within 48 h were included. Later opt-in consent was sought for body composition imaging outcomes. Trial data were obtained from the UK National Neonatal Research Database (NNRD), supplemented with prospectively recorded daily nutritional intake data until discharge. In addition, the feeding mode was recorded at the scan visit as either “breast”, “formula”, or “mixed”. Infants were fed MM or other milk if MM was unavailable prior to randomisation at the discretion of the attending clinician. Randomisation was to one of three groups: (i) unfortified MM supplemented with unfortified DM (UHM); (ii) fortified MM supplemented with fortified DM (FHM); (iii) unfortified MM supplemented with preterm formula (PTF). Preterm formula was Cow and Gate Nutriprem 1 (Nutricia Ltd., Newmarket Avenue, White Horse Business Park, Trowbridge, Wiltshire, UK; 80 kcal/100 mL energy, 2.6 g/100 mL protein, 3.9 g/100 mL fat, 8.4 g/100 mL carbohydrate). Fortifier (Cow and Gate Nutriprem; 15 kcal/100 mL energy, 1.1 g/100 mL protein, 0 g/100 mL fat, 2.7 g/100 mL carbohydrate) was introduced in the FHM group once a total enteral feed volume of 100 mL/kg/d was reached. Feeding intervention continued until 35 + 0 weeks PMA, after which they were transitioned to suck feeds by breast or bottle according to maternal choice with continuation of any fortification at the discretion of the attending clinician.

Pasteurised (Holder method) donor milk was supplied by the North-West UK Human Milk Bank, which is routinely analysed for macronutrient content prepasteurisation [18]. Milk volumes were increased to a maximum of 200 mL/kg/d for human milk and 165 mL/kg/d for PTF, as per recommendation, modified in accordance with infant tolerance, and parenteral nutrition was weaned in accordance with standard guidelines. If an infant required cessation of enteral feeds, the randomised regimen was resumed upon recommencement unless considered inappropriate by the attending clinical team. All infants were included in the intention-to-treat analysis unless withdrawn.

### 2.2. Study Outcomes

Details of trial feasibility, anthropometry, and body composition outcomes are reported elsewhere [17]. A preplanned secondary comparison of term anthropometry and body composition, as measured by whole-body MRI [11], in accordance with actual predominant milk intake was undertaken. Four mutually exclusive groups were defined based on the proportion of intake of human milk or preterm formula from birth to 35 weeks PMA: exclusive UHM (ExUHM, proportion of UHM 99–100%), predominantly UHM (PrUHM, proportion of UHM 50–98.9%), predominantly FHM (PrFHM, proportion of FHM > 50%), and predominantly preterm formula (PrPTF, proportion of PTF > 50%). Comparisons between groups of the change in anthropometry z scores from birth to end of feeding intervention at 35 + 0 weeks PMA and to discharge as well as macronutrient intake during total hospital stay using reference values for MM [19] and prepasteurisation analysis for DM were undertaken to aid interpretation of the anthropometry and body composition outcomes. Infants who had all milk intake from suckling at the breast at or shortly after 35 weeks PMA were excluded from the nutritional analysis from 35 weeks PMA to discharge.

For body composition imaging, we scanned infants supine, in natural postprandial sleep, without sedation in accordance with our well-established protocol [11]. Data were acquired on a 1.5 T (52 infants) or 3.0 T MRI scanner (12 infants). We used a T1-weighted fast spin echo axial sequence with a repetition time of 514 ms (500 ms 3.0 T), echo time of 11 ms (20 ms 3.0 T), echo train length of 3, and 3 signal averages (3 and 1, 3.0 T). Each slice was 5 mm thick with a 5 mm gap. Total Adipose Tissue (TAT) volume was quantified as the sum of six discrete depots (Appendix A): Superficial Subcutaneous Abdominal Adipose Tissue (SSCAAT), Superficial Subcutaneous Non-Abdominal Adipose Tissue (SSNAAT), Deep Subcutaneous Abdominal Adipose Tissue (DSCAAT), Deep Subcutaneous Non-Abdominal Adipose Tissue (DSCNAAT), Internal Abdominal Adipose Tissue (IAAT), Internal Non-Abdominal Adipose Tissue (INAAT), as previously described [11]. Image analysis (Slice-OMatic V.4.2; Tomovision) was undertaken independently by Vardis Group, London, UK, blinded to participant identity and feeding group allocation. AT volume in litres was converted to AT mass in grams (g), assuming a value for the density of AT of 900 g/L. Non-ATM was calculated as (body weight (g) − (AT volume (cm^3^) × 0.9)).

### 2.3. Statistical Analysis

All analyses were performed with SPSS version 27. Statistical significance was defined as *p* < 0.05. Baseline and nutritional differences between feed groups were assessed using ANOVA with examination of the six pairwise differences. To allow for meaningful comparisons of relative adiposity, we adjusted for body size using ponderal index [20], age at scan, and sex [21]. We used multiple linear regression for adjusted comparisons between groups with additional covariates including birth gestation, birthweight z score, and level of care [22]. We plotted histograms for each outcome and checked standardised residuals for normality, with natural log transformation of the dependent variable for non-normality. Linear regression models were carried out using the enter method but also using the stepwise method to check that the coefficient of determination did not fall after adding each independent variable. We report the adjusted mean difference with 95% confidence intervals. All analyses were performed with the intention to treat. Sensitivity analyses were undertaken by excluding outcome outliers (>mean ± 3SD).

## 3. Results

### 3.1. Baseline Characteristics

Of the original 103 infants randomised, 2 died and 6 withdrew from the feed intervention. Of the 95 surviving infants who continued in this study, 62 out of 64 that completed the body composition primary outcome had complete information on milk intake until 35 weeks PMA, 23 of which were exclusively (≥99%) UHM fed (ExUHM), 15 were predominantly (>50% <99%) UHM fed (PrUHM), 17 were predominantly (>50%) FHM fed (PrFHM), and 7 were predominantly (>50%) PTF fed. There were no infants who fell outside of these categories. Baseline characteristics, anthropometry, and body composition data by feed exposure are presented in Table 1. Baseline characteristics including birth gestation, birthweight, and birthweight z score were generally well balanced between groups, but there was a predominance of boys in the PrPTF group. Median (IQR) gestational age and birthweight overall was 30 + 0 (28 + 5 to 31 + 1) weeks and 1.3 (1.1 to 1.6) kg, respectively.

### 3.2. Milk and Macronutrient Intake

Milk and macronutrient intake for feed exposure groups are shown in Table 2. The length of standard parenteral nutrition did not significantly differ between groups. In-hospital DM (fortified or unfortified) intake was low for all infants, with an overall mean (SD) daily intake of 13.0 (25.3) mL/kg/d. Although some infants had started to breastfeed prior to 35 weeks PMA, this appeared to have little impact on the volume of milk intake by nasogastric tube, orogastric tube, or bottle (overall median (IQR) of the mean daily milk intake from 35 weeks PMA to discharge: 162.4 (134.1 to 173.0) mL/kg/d). The mean daily volume of milk intake (mL/kg/d) by these routes did not significantly differ between groups for the duration of feed intervention nor from 35 weeks PMA till discharge. From 35 weeks PMA to discharge, PrPTF intake of formula continued to be significantly higher than all other groups. Although the majority of infants discontinued fortifier at 35 weeks PMA, 6 out of 17 continued fortification for a further 1–2 weeks, with 2 infants still on fortifier at 37 + 0 weeks; there were no infants receiving fortified MM at discharge. Overall, there was a trend from breast milk feeding at discharge to formula feeding by scan visit. Median (IQR) PMA at discharge overall was 37 + 2 (35 + 5 to 39 + 2) weeks.

From birth to 35 weeks PMA, PrFHM average daily protein, energy, carbohydrate, and protein/energy ratio were significantly higher than in any other group, while the PrPTF group also had a significantly higher average daily protein/energy ratio intake than the ExUHM and PrUHM groups. However, there were no significant pairwise differences between groups in macronutrient intake from 35 weeks PMA till discharge.

### 3.3. Term Body Composition

Median age (IQR) at term scan was 40 + 6 (39 + 3 to 42 + 0) weeks PMA. In comparison to the ExUHM group, PrPTF group had (mean difference (95% CI)) 274.3g (30.1 to 518.5 g) more NATM in the fully adjusted model and a 35% (5.1 to 73.2%) significantly higher IAAT (Table 3). However, there did not appear to be any disproportionate adiposity in any particular depot and no significant differences in percentage depot adiposity (of total adiposity) between groups (see Appendix A). There were no other significant differences in any body composition measure between PrPTF and the PrFHM and PrUHM groups.

### 3.4. Anthropometry

In the fully adjusted models, in comparison to the ExUHM group, the PrPTF fed group had consistently significantly higher weight gain at the end of the feeding intervention, at discharge, and at term (term weight z score change 1.11 (0.38 to 1.84)), while the PrFHM group had significantly higher weight gain at discharge only. The PrPTF and PrFHM groups also had significantly higher length gain at term in comparison to the ExUHM group (PrPTF: length z score change 1.22 (0.18 to 2.26); PrFHM: length z score change 0.87 (0.08 to 1.67)).

The PrPTF group also had significantly higher weight gain than the PrUHM group at the end of the feeding intervention and both the PrUHM and PrFHM groups at discharge (PrPTF vs. PrFHM weight z score change to discharge: 0.69 (0.13 to 1.24), *p* = 0.02), but there were no other significant differences in anthropometry between the PrPTF, PrFHM, and PrUHM groups at all timepoints.

### 3.5. Sensitivity Analyses

In all regression models, the coefficients of determination were well above 0.4, suggesting a large effect, and using the stepwise method as a check, the coefficient of determination did not fall after adding each independent variable. Sensitivity analyses involving regression analyses excluding outliers for each outcome (body composition: INAAT × 1; DSCAAT × 1; DSCNAAT × 1; anthropometry: weight z score change birth to 35 weeks × 1; weight z score change to discharge × 1; head circumference z score change birth to 35 weeks × 2; head circumference z score change to discharge × 1) did not change the results.

## 4. Discussion

In this secondary analysis of the PREMFOOD study [17], PrPTF-fed infants had significantly higher NATM, weight, and length z score gains than ExUHM-fed infants at term. PrFHM-fed infants also had significantly higher length but not weight z score gains in comparison to ExUHM-fed infants.

Unravelling the effect of PTF in comparison to HM feeding on very preterm body composition has been complicated due to indirect body composition measurement methods and inadequate separation of the effect from bovine fortification of HM. The strengths of this study lie in the body composition evaluated using a direct gold-standard approach [23], with the ability to distinguish total and regional adipose depots, analyses of imaging data blind to feed exposure, and detailed data on nutritional intake across the hospital stay from the original RCT, comparing UHM, FHM, and PTF, three widely used feeding regimens globally.

This study has several limitations. Many parents declined the imaging studies, leading to a reduction of power for the anthropometry and body composition outcomes, and the results need to be interpreted with caution, especially considering the small sample size of the PrPTF group. Although not suggested by statistical checks, the small sample size also raises the risk of overfitting the body composition/anthropometry models with the covariates used. Illness severity is a known determinant of body composition at term in very preterm infants [24]. The number of level 1 and level 2 care days as a percentage of the total stay was used as a proxy of illness severity and as a covariate in the regression models. It is possible, though, that this did not capture illness severity adequately or that body composition was affected by other unmeasured confounders.

At 35 weeks PMA, infants were transitioned to suck feeds, and self-regulated intake between 35 weeks PMA and term may impact body composition. It is not possible to assess breastfeeding milk intake, and detailed nutritional intake data were not collected for the time period from discharge to term scan. However, there were no significant differences in daily milk intake volume by bottle or naso/orogastric tube between feed exposure groups from 35 weeks PMA till discharge, and all study infants continued to have average daily intakes by these routes well above 100 mL/kg/d during this period, with discharge home soon after full breastfeeding or bottle feeding was established.

Our findings of higher NATM in those infants predominantly formula-fed compared to exclusively UHM-fed are consistent with those of Li et al. [16], who found higher NATM, also measured by MRI, in PrPTF- compared to exclusively HM-fed (unfortified or fortified) very preterm infants until 34 weeks PMA. Contrary to these findings, in a review of observational studies of preterm feeding, Cerasani et al. [25] reported that predominant or exclusive FHM feeding in comparison with predominant or exclusive PTF feeding is associated with lower fat mass and fat mass percent and higher fat-free mass percent at term. This discrepancy may be explained by the fact that in many of the observational studies from this review, fortified HM was continued until discharge or term corrected age rather than until 34–35 weeks PMA.

In contrast to Li et al., who found no difference, we also found higher IAAT in PrPTF compared to ExUHM fed infants. Higher protein intake in formula fed healthy term infants was shown to disproportionately increase preperitoneal or visceral fat tissue at the age of 5 years [26]. However, we did not find any evidence of disproportionate depot adiposity, as there were no differences between ratios of depot adipose tissue to total adipose tissue between groups. This suggests that nutrition does not influence the adverse adipose partitioning seen in very preterm infants at term compared to term-born controls [11], although a longer-term programming effect cannot be excluded; follow-up body composition studies of this cohort in childhood are currently underway.

Timing of milk and nutrient intake raises intriguing insights into the differential effect on very preterm body composition at term. With sufficient energy intake, increasing protein intake is associated with lean mass accretion [27,28]. Interestingly, despite the highest protein/energy ratio intake in the PrFHM group until 35 weeks PMA, NATM accretion was highest in the PrPTF group at term. Although DM was analysed for macronutrient content, intake was low across all human milk groups, and we used reference values for MM macronutrient content [20]. Due to the variable nutrient content of human milk [29,30], it is possible that these were overestimated. Alternatively, or in addition, higher NATM at term seen in the PrPTF group could simply represent higher cumulative protein/energy intakes with persistent formula diet until term: only a third of infants in the FHM group continued fortification beyond 35 weeks PMA, with none on fortifier at discharge, and there was a trend to increasing formula intake in the PrPTF group from discharge to term scan, with none exclusively HM fed at these timepoints. This is consistent with higher fat-free mass at term seen in exclusively formula fed preterm infants randomised to nutrient enriched formula compared to term formula at discharge or compared to breastfeeding controls [31]. Furthermore, in a systematic review of body composition studies of formula versus breastfed term infants, formula feeding was associated with higher fat-free mass than breastfed infants until 7 months of age [32].

The effect of very early nutrient provision on body composition at term, however, is less clear. Observational studies reporting increased fat-free mass [33,34], measured by air displacement plethysmography (ADP) during hospital stay in very low birthweight infants, to be associated with high protein intakes in the first week of age are not supported by trial data in very preterm infants, reporting no effect of increased parenteral protein in the first few days of life on MRI-measured non-adipose mass at term [35]. Similarly, in a recent trial of extremely preterm infants randomised to early FHM (in which fortification was human-milk-derived) versus early UHM, before all infants started FHM diets (fortification was bovine-derived) once full feeds were established, there were no differences in fat-free mass measured by ADP at term age, although the early FHM group had higher gains in length and head circumference z scores than the early UHM group [36].

## 5. Conclusions

In summary, we have shown increased NATM, weight, and length z score gains in very preterm infants at term fed a PrPTF diet in comparison to an ExUHM diet. Despite higher gains in length z score between birth and term, these gains in NATM were not matched by early routine fortification of HM. These differences may relate to the persistently longer PTF diet and therefore cumulatively higher macronutrient intake of PrPTF fed infants. Adequately powered very preterm feeding clinical trials for both short and long term health outcomes are required to reduce uncertainty in optimal supplemental feeding; as part of these trials, whether anthropometry and body composition at term are altered from prolonged or delayed fortification strategies are important research questions.

## Figures and Tables

**Table 1 nutrients-17-01366-t001:** Demographic and clinical characteristics by feed exposure group.

Demographic/Clinical Characteristic *	ExUHM (n = 23)	PrPTF (n = 7)	PrFHM (n = 17)	PrUHM (n = 15)	*p*-Value ^$^
Sex (%)					0.49
Male	52.2	85.7	58.8	53.3	
Female	47.8	14.3	41.2	46.7	
Gestational age (weeks)	30.0 (28.7 to 31.0)	29.3 (26.3 to 31.7)	30.3 (28.9 to 30.7)	30.3 (28.1 to 31.6)	0.93
Birth weight (g)	1235 (1077 to 1643)	1436 (880 to 1567)	1313 (1225 to 1555)	1333 (1070 to 1540)	0.93
Birth weight z score	−0.15 (−1.15 to 0.46)	−0.34 (−0.85 to 0.22)	−0.55 (−1.08 to 0.00)	−0.22 (−0.89 to 0.06)	0.86
Birth length (cm)	39.0 (37.2 to 41.4)	37.5 (34.2 to 41.0)	38.6 (37.0 to 40.0)	38.5 (37.2 to 40.8)	0.63
Birth OFC (cm)	27.9 (25.6 to 29.0)	28.5 (25.0 to 29.6)	27.0 (26.0 to 28.7)	27.5 (26.0 to 29.0)	0.96
Multiple birth (%)					0.78
No	52.2	71.4	47.1	46.7	
Yes	47.8	28.6	52.9	53.3	
Small for gestational age (<1.28 z score) (%)	17.4	14.3	17.6	7.0	0.88
Received antenatal steroids (%)	91.3	100	88.2	100	0.65
Apgar score at 5 min	8.0 (8.0 to 10.0)	8.0 (7.0 to 10.0)	9.0 (8.0 to 9.0)	9.0 (9.0 to 9.0)	0.93
Maternal age years	35.0 (33.0 to 38.0)	35.0 (28.0 to 35.0)	35.0 (33.0 to 38.0)	37.0 (31.0 to 38.0)	0.38
Maternal parity (%)					0.39
1	65.2	71.4	82.3	86.7	
>1	34.8	28.6	17.7	13.3	
Maternal ethnicity (%)					0.76
White	60.9	85.7	76.5	79.9	
Mixed	8.7	14.3	5.9	6.7	
Asian	8.7	0	0	6.7	
Black	13.0	0	17.6	6.7	
Other	8.7	0	0	0	
Percent level 1 and 2 care ^#^	46.1 (22.2 to 63.0)	52.4 (25.8 to 80.4)	27.0 (18.3 to 64.7)	49.4 (11.1 to 61.3)	0.78
35 weeks PMA					
Weight (g)	1846 (1575 to 2170)	2022 (2030 to 2560)	1962 (1760 to 2120)	2022 (1747 to 2215)	0.48
Length (cm)	43.5 (41.0 to 45.6)	44.0 (43.1 to 44.9)	43.7 (41.1 to 44.7)	43.9 (42.5 to 44.6)	0.82
Head circumference (cm)	30.8 (29.4 to 32.0)	31.7 (31.0 to 32.0)	30.8 (29.9 to 32.0)	31.2 (30.8 to 32.0)	0.80
Term					
Weight (g)	2875 (2260 to 3510)	4000 (3180 to 4040)	3220 (2940 to 3760)	3280 (3040 to 3720)	0.05
Length (cm)	43.5 (45.0 to 52.5)	51.0 (50.0 to 55.0)	50.5 (47.5 to 52.0)	50.0 (48.0 to 52.0)	0.18
OFC (cm)	35.3 (32.8 to 36.4)	37.0 (35.8 to 37.8)	35.6 (34.1 to 36.5)	35.8 (34.7 to 37.4)	0.13
Term					
TAT (L)	0.761 (0.492 to 0.974)	1.138 (0.794 to 1.271)	0.968 (0.677 to 1.102)	0.818 (0.710 to 0.971)	0.11
NATM (g)	2306.5 (1771.3 to 2593.3)	2876.1 (2465.4 to 2975.8)	2441.7 (2188.2 to 2769.4)	2543.8 (2209.5 to2949.9)	0.03
Percent adiposity	23.5 (18.9 to 25.4)	25.6 (22.5 to 28.5)	25.2 (20.7 to 28.6)	23.0 (21.0 to 24.3)	0.20

* Data presented are median (IQR) for continuous variables and frequency (percentage) for categorical variables; ^#^ defined as number of days in level 1 and level 2 care as a percentage of total days in hospital [23]; ^$^ ANOVA for continuous variables, Fisher’s exact test for categorical variables; OFC, Occipital Frontal Diameter; PMA, postmenstrual age.

**Table 2 nutrients-17-01366-t002:** Average daily milk and total macronutrient (parenteral and enteral ^#^) intake mean (sd) between birth and 35 weeks PMA (end of feed intervention) (T1); between 35 weeks PMA and discharge (T2); and discharge and scan feed type percent by feed exposure group from birth to 35 weeks PMA.

Outcome	ExUHM (a) n = 23	PrPTF (b) n = 7	PrFHM (c) n = 17	PrUHM (d) n = 15	*p*-Value; Pairwise Comparisons ^$^
Days on PN	10.9 (3.5)	13.4 (4.8)	13.2 (8.3)	11.7 (7.9)	0.63
UDM mL/kg/d (T1)	11.6 (26.3)	17.2 (29.8)	3.3 (5.7)	21.4 (39.4)	0.30
UDM mL/kg/d (T2)	0.9 (3.9) n = 20	0 (0)	0 (0) n = 16	0 (0)	0.63
FDM mL/kg/d (T1)	0 (0)	0 (0)	14.7 (26.5)	0 (0)	0.01; c > a **; c > d *
FDM mL/kg/d (T2)	0 (0) n = 20	0 (0)	0.8 (3.1) n = 16	0 (0)	0.47
UHM mL/kg/d (T1)	135.8 (18.3)	41.3 (29.5)	20.1 (11.9)	102.8 (38.5)	<0.001; a > b; a > c; a > d; d > b; d > c
UHM mL/kg/d (T2)	123.0 (45.0) n = 20	13.0 (22.7)	79.9 (63.4) n = 16	85.1 (69.0)	<0.001; a > b; d > b *
FHM mL/kg/d (T1)	0 (0)	0 (0)	110.6 (27.7)	10.2 (19.1)	<0.001; c > a; c > b; c > d
FHM mL/kg/d (T2)	6.1 (27.2) n = 20	0 (0)	34.1 (44.3) n = 16	14.3 (53.5)	0.13
Form mL/kg/d (T1)	0.3 (0.4)	78.2 (32.2)	2.5 (6.5)	16.2 (22.2)	<0.001; b > a; b > c; b > d; d > a *
Form mL/kg/d (T2)	18.8 (36.6) n = 20	133.9 (52.0)	30.9 (54.0) n = 16	50.9 (55.3)	<0.001; b > a; b > c; b > d **
Total milk mL/kg/d (T1)	136.1 (18.3)	119.5 (21.4)	133.2 (21.8)	129.2 (29.2)	0.38
Total milk mL/kg/d (T2)	147.8 (37.5) n = 20	146.9 (33.1)	145.0 (51.9) n = 16	150.3 (52.9)	0.99
Protein g/kg/d (T1)	2.68 (0.24)	3.01 (0.57)	3.84 (0.40)	2.75 (0.51)	<0.001; c > a; c > b; c > d
Protein g/kg/d (T2)	2.53 (0.82) n = 20	2.98 (0.68)	2.91 (0.80) n = 16	3.14 (0.90)	0.20
Energy kcal/kg/d (T1)	110.78 (8.57)	107.39 (11.83)	124.15 (10.60)	109.95 (12.02)	<0.001; c > a; c > b **; c > d **
Energy kcal/kg/d (T2)	100.93 (26.68) n = 20	109.29 (25.79)	108.16 (26.63) n = 16	118.09 (22.14)	0.44
Pro:En g/100 kcal (T1)	2.42 (0.13)	2.79 (0.30)	3.09 (0.12)	2.50 (0.32)	<0.001; c > a; c > d; c > b *; b > a *; b > d *
Pro:En g/100 kcal (T2)	2.47 (0.19) n = 20	2.74 (0.12)	2.67 (0.30) n = 16	2.60 (0.32)	0.04; b > a ***; c > a ***
Fat g/kg/d (T1)	5.28 (0.48)	4.95 (0.71)	5.14 (0.56)	5.07 (0.76)	0.56
Fat g/kg/d (T2)	4.66 (1.94) n = 20	5.65 (1.42)	5.09 (1.84) n = 16	5.49 (1.81)	0.40
CHO g/kg/d (T1)	13.04 (1.19)	12.45 (0.87)	15.58 (1.24)	13.26 (1.17)	<0.001; c > a; c > b; c > d
CHO g/kg/d (T2)	9.99 (4.18) n = 20	11.28 (2.48)	11.53 (4.18) n = 16	12.02 (4.33)	0.35
Feed at discharge (%)					<0.001
Breast milk	82.6	0	64.7	46.7	
Formula	0	57.14	29.4	33.3	
Mixed	17.4	42.6	5.9	20.0	
Feed at term scan (%)					<0.001
Breast milk	60.9	0	52.9	26.7	
Formula	8.7	85.7	11.8	26.7	
Mixed	30.4	14.3	35.3	46.6	

^#^ Enteral intake defined as all nasogastric/orogastric or bottle milk feeds—i.e., breastfeeding excluded. ^$^ Between group ANOVA and pairwise post hoc Tukey HSD tests shown with *p* < 0.1 for continuous variables, all pairwise *p* values <0.001 except * *p* < 0.05 and ≥ 0.01; ** *p* < 0.01 and ≥ 0.001; and *** *p* < 0.1 and ≥ 0.05; Fisher’s exact test for categorical variables; UDM, unfortified pasteurised human donor milk (DM); FDM, fortified DM; UHM, unfortified human milk (maternal milk (MM) and/or DM); FHM, fortified human milk; Form, formula (preterm or other); Pro: En, protein/energy ratio; CHO, carbohydrate.

**Table 3 nutrients-17-01366-t003:** Regression analyses for anthropometry and body composition at term by feed exposure group.

		ExUHM n * = 23	PrPTF n * = 7	PrFHM n * = 17	PrUHM n * = 15
∆Wt z score	Unadj	Ref	0.77 (0.28 to 1.25), *p* <0.01	0.38 (0.03 to 0.74), *p* = 0.04	0.22 (−0.15 to 0.59), *p* = 0.24
birth to 35w	Adj ^1^	Ref	0.75 (0.30 to 1.21), *p* < 0.01	0.32 (−0.02 to 0.66), *p* = 0.06	0.18 (−0.17 to 0.66), *p* = 0.06
∆Wt z score	Unadj	Ref	1.11 (0.57 to 1.66), *p* < 0.001	0.47 (0.07 to 0.87), *p* = 0.02	0.28 (−0.14 to 0.70), *p* = 0.19
birth to disch	Adj ^2^	Ref	1.09 (0.56 to 1.62), *p* < 0.001,	0.40 (0.01 to 0.80), *p* = 0.04	0.25 (−0.15 to 0.66), *p* = 0.22
∆Wt z score	Unadj	Ref	1.12 (0.40 to 1.84), *p* < 0.01	0.52 (−0.01 to 1.06), *p* = 0.05	0.34 (−0.22 to 0.89), *p* = 0.23
birth to term	Adj ^1^	Ref	1.11 (0.38 to 1.84), *p* < 0.01	0.49 (−0.05 to 1.03), *p* = 0.08	0.32 (−0.24 to 0.88), *p* = 0.25
∆Length z score	Unadj	Ref	0.52 (−0.08 to 1.11), *p* = 0.09	0.21 (−0.23 to 0.65), *p* = 0.34	0.02 (−0.45 to 0.48), *p* = 0.95
birth to 35w	Adj ^1^	Ref	0.51 (−0.05 to 1.07), *p* = 0.08	0.15 (−0.27 to 0.56), *p* = 0.48	−0.03 (−0.47 to 0.41), *p* = 0.90
∆Length z score	Unadj	Ref	1.18 (0.14 to 2.22), *p* = 0.03	0.81 (0.02 to 1.60), *p* = 0.04	−0.19 (−1.03 to 0.65), *p* = 0.65
birth to term	Adj ^1^	Ref	1.22 (0.18 to 2.26), *p* = 0.02	0.87 (0.08 to 1.67), *p* = 0.03	−0.12 (−0.96 to 0.72), *p* = 0.77
∆OFC z score	Unadj	Ref	0.64 (−0.49 to 1.76), *p* = 0.26	0.16 (−0.71 to 1.02), *p* = 0.72	0.02 (−0.86 to 0.91), *p* = 0.96
birth to term	Adj ^1^	Ref	0.54 (−0.58 to 1.67), *p* = 0.34	0.07 (−0.80 to 0.94), *p* = 0.87	−0.03 (−0.90 to 0.85), *p* = 0.96
TAT (L)	Unadj	Ref	0.293 (0.037 to 0.548), *p* = 0.03	0.162 (−0.027 to 0.351), *p* = 0.09	0.081 (−0.115 to 0.277), *p* = 0.41
	Adj ^3^	Ref	0.165 (−0.009 to 0.338), *p* = 0.06	0.080 (−0.047 to 0.207), *p* = 0.21	−0.081 (−0.218 to 0.057), *p* = 0.25
Non-ATM (g)	Unadj	Ref	506.2 (119.3 to 893.0), *p* = 0.01	224.5 (−62.1 to 511.1), *p* = 0.12	340.6 (43.2 to 638.0), *p* = 0.03
	Adj ^3^	Ref	274.3 (30.1 to 518.5), *p* = 0.03	136.5 (−42.0 to 315.1), *p* = 0.13	159.6 (−34.4 to 353.6), *p* = 0.11
% ATM	Unadj	Ref	2.9 (−0.7 to 6.4), *p* = 0.11	2.1 (−0.5 to 4.7), *p* = 0.12	0.02 (−2.7 to 2.7), *p* = 0.99
	Adj ^4^	Ref	2.3 (−1.0 to 5.7), *p* = 0.17	1.7 (−0.7 to 4.1), *p* = 0.16	−1.2 (−3.8 to 1.4), *p* = 0.36
IAAT	Unadj	Ref	49.0 (6.7 to 108.1), *p* = 0.02	24.5 (−2.9 to 59.4), *p* = 0.08	17.2 (−9.3 to 51.6), *p* = 0.22
(% change)	Adj ^3^	Ref	35.0 (5.1 to 73.2), *p* = 0.02	15.0 (−4.0 to 38.3), *p* = 0.13	0.1 (−17.9 to 22.0), *p* = 0.99
INAAT	Unadj	Ref	41.1 (−0.3 to 99.4), *p* = 0.05	9.9 (−15.1 to 42.1), *p* = 0.47	16.0 (−11.2 to 51.5), *p* = 0.27
(% change)	Adj ^3^	Ref	19.1 (−8.7 to 55.4), *p* = 0.19	−1.1 (−18.5 to 20.2), *p* = 0.91	−5.1 (−23.1 to 17.4), *p* = 0.63
DSCAAT	Unadj	Ref	66.2 (1.1 to 173.2), *p* = 0.05	26.1 (−12.7 to 82.2), *p* = 0.21	12.9 (−23.0 to 65.4), *p* = 0.53
(% change)	Adj ^3^	Ref	31.7 (−6.8 to 86.1), *p* = 0.12	7.3 (−16.6 to 38.1), *p* = 0.58	−18.5 (−38.1 to 7.3), *p* = 0.14
DSCNAAT	Unadj	Ref	47.3 (−0.1 to 117.1), *p* = 0.05	33.8 (0.3 to 78.4), *p* = 0.05	26.0 (−6.5 to 69.9), *p* = 0.13
(% change)	Adj ^3^	Ref	25.7 (−9.4 to 74.5), *p* = 0.17	23.6 (−2.8 to 57.1), *p* = 0.08	6.9 (−17.6 to 38.8), *p* = 0.61
SSCAAT (L)	Unadj	Ref	0.055 (0.008 to 0.103), *p* = 0.02	0.023 (−0.012 to 0.059), *p* = 0.20	0.005 (−0.032 to 0.042), *p* = 0.78
	Adj ^3^	Ref	0.031 (−0.006 to 0.067), *p* = 0.10	0.005 (−0.021 to 0.032), *p* = 0.68	−0.028 (−0.057 to 0.001), *p* = 0.06
SSCNAAT (L)	Unadj	Ref	0.191 (0.023 to 0.360), *p* = 0.03	0.117 (−0.008 to 0.242), *p* = 0.07	0.063 (−0.067 to 0.1920, *p* = 0.34
	Adj ^3^	Ref	0.110 (−0.008 to 0.227), *p* = 0.07	0.066 (−0.020 to 0.152), *p* = 0.13	−0.040 (−0.133 to 0.054), *p* = 0.40

* Number of infants with detailed nutritional intake available; Unadj, unadjusted; Adj, adjusted; ∆, change. ^1^ Adjusted for birth gestational age, birthweight z score, and percent levels 1 and 2 care days of total stay; ^2^ adjusted for model 1 covariates and gestational age at discharge; ^3^ adjusted as for model 1 covariates, sex, age at scan, and ponderal index at scan (Wt (kg)/length (m)^3^); ^4^ adjusted for model 1 covariates, sex, and age at scan.

## Data Availability

The raw data supporting the conclusions of this article will be made available by the authors on request.

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
