# Peer review of "Impact of Unfortified Human Milk, Fortified Human Milk, and Preterm Formula Intake on Body Composition at Term in Very Preterm Infants: Secondary Analysis of the PREMFOOD Trial"

_nutrients, 2025, doi:10.3390/nu17081366_

Round 1
Reviewer 1 Report
Comments and Suggestions for Authors
This manuscript explores the impact of in-hospital exclusive unfortified human milk (HM), predominant unfortified HM, predominant fortified HM, and predominant preterm formula on very preterm infant body composition and anthropometry at term (35 weeks postmenstrual age), discharge, and post-discharge scan. Prior studies were mixed on the effects of feeding strategies on infant adiposity and fat free mass. In this study, infants fed predominantly formula had a greater increase in non-adipose tissue mass and weight z-scores compared to exclusively HM-fed infants, while predominantly formula and fortified HM infants had greater increases in length z-scores.
Major:
1. The experimental groups are a bit confusing. The description in the Methods reads well, but that in the Abstract and end of Introduction is a bit confusing. Suggest rewording the Abstract and Introduction to better match that of the Methods. A visual scheme of the groups would also help to clarify the experimental design.
2. Please provide statistics for all metrics reported in Table 1. Providing ANOVA p-values for all metrics in Table 2, not just multiple comparisons that were significant, would also be more transparent.
3. Could the authors add to the Discussion the clinical implications for any recommendations they would make based upon this data?
Minor:
1. In abstract, Line 24, “weight” is misspelled.
2. Lines 104-107 state that infants were included in the study even if enteral feeding was stopped at some point. Wouldn’t gaps in feeding make results difficult to interpret?
Author Response
Comment 1
The experimental groups are a bit confusing. The description in the Methods reads well, but that in the Abstract and end of Introduction is a bit confusing. Suggest rewording the Abstract and Introduction to better match that of the Methods. A visual scheme of the groups would also help to clarify the experimental design.
Response 1
Thank you for your comments. Abstract and end of introduction has been reworded to better match that of the methods. Please see revised manuscript. A visual scheme of the groups has been carefully considered, but we did not think this would add to the clarity of the experimental design.
Comment 2
Please provide statistics for all metrics reported in Table 1. Providing ANOVA p-values for all metrics in Table 2, not just multiple comparisons that were significant, would also be more transparent.
Response 2
Statistics have been provided for table 1 and table 2.
Comment 3
Could the authors add to the Discussion the clinical implications for any recommendations they would make based upon this data?
Response 3
The only recommendations, as stated, would be for further larger studies comparing different fortification strategies, and preterm formula to human donor milk supplement, the clinical implications of which remain to be determined.
Comment 4
In abstract, Line 24, “weight” is misspelled.
Response 4
This has been corrected.
Comment 5
Lines 104-107 state that infants were included in the study even if enteral feeding was stopped at some point. Wouldn’t gaps in feeding make results difficult to interpret?
Response 5
This is a valid point. The original trial was designed as a pragmatic feasibility study, however the number of days on parenteral nutrition was not significantly different between groups, and there were no outliers for each group.
Reviewer 2 Report
Comments and Suggestions for Authors
This is a manuscript reporting about a pre-planned secondary analysis of the PREMFOOD trial, assessing body composition at term equivalent age in very preterm infants randomized to different feeding regimens. The topic is of great importance and interesting for both, clinicians having to decide about nutrition of their NICU patients on a daily basis, as well as for researchers in this area.
The manuscript is very well structured and written and it was a pleasure to read it.
Some minor comments
Abstract, results part: typo “wieght” instead of weight
There are quite a lot of abbreviations used in the text (for different milk types; groups based on predominance, different sites of non-adipose tissue mass measurement) which can make it a bit hard for the reader to keep the flow.
Introduction:
- “Using magnet resonance imagine (MRI), our group has (instead of have) previously shown…”
Methods:
- It would be very interesting to add maternal BMI (preferably before pregnancy) as potential predictor, if we assume a genetic disposition.
- Did you consider looking at medication the infants were on, e.g. diuretics, postnatal steroids?
- Would it be possible to look at impact of timing of initiation of enteral feeds (days of life)?
Results:
- mLs/kg/d should be mL/kg/d (in Table 2 and results part 3.2. milk and micronutrient intake), at most other sections written without “s”
- Table 1: is the maternal age in the PrPTF group correct (as median of 35.0 seems a lot more skewed than in the other groups, if the IQR is 28.0-35.0)
Author Response
Comment 1
Abstract, results part: typo “wieght” instead of weight
Response 1
Thank you for your comments. Weight has been corrected. Please see revised manuscript.
Comment 2
There are quite a lot of abbreviations used in the text (for different milk types; groups based on predominance, different sites of non-adipose tissue mass measurement) which can make it a bit hard for the reader to keep the flow.
Response 2
We appreciate there are lots of abbreviations, however we feel this is necessary so that the reader knows exactly what type of milk and feed group is being referred to. We have provided a figure to refer to the different adipose compartments in the supplementary information.
Comment 3
Introduction:
“Using magnet resonance imagine (MRI), our group has (instead of have) previously shown…”
Response 3
This has been corrected.
Comment 4
Methods:
It would be very interesting to add maternal BMI (preferably before pregnancy) as potential predictor, if we assume a genetic disposition.
Response 4
This was looked at. Data was collected for booking maternal BMI, not pre-pregnancy BMI. Unfortunately, there was some missing data, so that final numbers were only 56 in the model including maternal BMI predictor. Due to small numbers we elected to not include this. However, inclusion of the predictor did not change the relationships.
Comment 5
Did you consider looking at medication the infants were on, e.g. diuretics, postnatal steroids?
Response 5
Data on diuretic use was not captured. There were no infants who received postnatal steroids (dexamethasone) for bronchopulmonary dysplasia, and only 3 who received hydrocortisone for hypotension, precluding any meaningful analysis.
Comment 6
Would it be possible to look at impact of timing of initiation of enteral feeds (days of life)?
Response 6
Data was collected on hrs of life milk first started. This was not significantly different between groups, and did not have any effect on the model results when included as a predictor.
Comment 7
Results:
mLs/kg/d should be mL/kg/d (in Table 2 and results part 3.2. milk and micronutrient intake), at most other sections written without “s”
Response 7
This has been corrected.
Comment 8
Table 1: is the maternal age in the PrPTF group correct (as median of 35.0 seems a lot more skewed than in the other groups, if the IQR is 28.0-35.0)
Response 8
This has been re-checked and is correct: Median 35; IQR 28-35; Range 18 (min 19 max 37)
Round 2
Reviewer 1 Report
Comments and Suggestions for Authors
The authors addressed most of my comments.
Author Response
N/A